# Integrated solid-state NMR and molecular dynamics modeling determines membrane insertion of human β-defensin analog

Xue Kang [1,3], Christopher Elson[2,3], Jackson Penfield[2], Alex Kirui [1], Adrian Chen[1], Liqun Zhang[2]* & Tuo Wang [1]*

Human β-defensins (hBD) play central roles in antimicrobial activities against various microorganisms and in immune-regulation. These peptides perturb phospholipid membranes for function, but it is not well understood how defensins approach, insert and finally disrupt membranes on the molecular level. Here we show that hBD-3 analogs interact with lipid bilayers through a conserved surface that is formed by two adjacent loops in the solution structure. By integrating a collection of $^{13}C$, $^{1}H$ and $^{31}P$ solid-state NMR methods with long-term molecular dynamic simulations, we reveal that membrane-binding rigidifies the peptide, enhances structural polymorphism, and promotes β-strand conformation. The peptide colocalizes with negatively charged lipids, confines the headgroup motion, and deforms membrane into smaller, ellipsoidal vesicles. This study designates the residue-specific, membrane-bound topology of hBD-3 analogs, serves as the basis for further elucidating the function-relevant structure and dynamics of other defensins, and facilitates the development of defensin-mimetic antibiotics, antifungals, and anti-inflammatories.

---

[1] Department of Chemistry, Louisiana State University, Baton Rouge, LA 70803, USA. [2] Department of Chemical Engineering, Tennessee Technological University, Cookeville, TN 38505, USA. [3] These authors contributed equally: Xue Kang, Christopher Elson. *email: lzhang@tntech.edu; tuowang@lsu.edu

Human β-defensins (hBD) are a family of antimicrobial peptides widely distributed in leukocytes and epithelial cells. These small cationic peptides (3–5 kDa) form the first line of immune defense against pathogenic infections by a broad spectrum of microorganisms, including Gram-positive and -negative bacteria, yeast, and encapsulated viruses[1–3]. Defensins also show chemotactic activity for monocytes, T cells and immature dendritic cells, reduce tissue damages by antimicrobial effectors[4–6], and inhibit tumor cell migration[7,8]. These biological functions, in particular, the microbicidal activity of defensins and other relevant antimicrobial peptides correlate with their capability of disrupting and permeabilizing membranes and causing membrane leakage to the invading microbes[9–13]. Efforts have been devoted to developing and engineering defensin-mimetic compounds as novel therapeutic agents against drug-resistant strains of bacteria and fungi, such as Candida albicans and Staphylococcus aureus[14,15], however, a major hurdle here is our limited understanding of the membrane-associated functional structure of defensins.

In humans, six α-defensins (~30 residues) and 28 types of β-defensins (33–47 residues) have been identified[16,17]. Both α- and β-defensins have six cysteine residues, but adopt distinct patterns in pairing and disulfide-bond formation. The solution or crystallographic structures of five β-defensins, including hBD-1, -2, -3, -4, and -6 have been determined[18–24], which feature a triple-stranded β-sheet fold as stabilized by three disulfide bonds. These peptides have a net charge density ranging from + 4 to + 11, and hBD-3, the focus of this study, is the most cationic peptide in this family, therefore, stronger interactions with the negatively charged bacterial membranes are expected[25]. Recently, the reduced analogs have also been shown to have comparable, if not greater, antimicrobial activity as the wild-type hBDs[26]. The analog adopts a structure that is less constrained than that of wild-type hBD-3, and retains the capability of disrupting the outer membrane of bacteria[27–29]. However, how these peptides interact with phospholipid membranes remain elusive and a central contribution of this study is to establish a high-resolution, residue-specific view of the insertion topology of hBD-3 analogs in lipid bilayers.

Three non-specific mechanisms of membrane-disruption have previously been proposed depending on the depth of insertion of the peptides. If only surface-coating occurs, the peptides will form a carpet outside the membrane and gradually dissolve the membrane like detergent[30]. When partially immersed in the membrane, the peptide may reside within one leaflet of the lipid bilayers and interrupt membrane integrity. If the peptide fully penetrates through the membrane, membrane-spanning pores can be formed as described in the toroidal and barrel-stave models, which are stabilized and supported by the oligomerization of peptides[31–34].

Here, we integrate 2D $^{13}$C–$^{13}$C, $^{13}$C–$^{1}$H, and $^{31}$P–$^{31}$P correlated solid-state NMR (ssNMR) spectroscopy with molecular dynamics (MD) simulation to provide the first site-specific evidence on the structure and dynamics of hBD-3 analogs in POPC/POPG lipid bilayers. Without the three disulfide bonds, the analog can still bind lipid bilayers using a conserved surface and efficiently break down the vesicles, which sheds light onto the membrane-disruption mechanism of this antimicrobial peptide. Membrane-binding enhances the structural polymorphism of hBD-3 analog, resulting in three conformers with distinct insertion patterns. The major conformer has a β-strand-dominant backbone conformation similar to that in solution, indicating an insignificant role of large-scale restructuring for function. This study provides novel insights into the functional structure of human β-defensins and membrane-disruption mechanism, which facilitates the development of defensin mimetics to address the increasing threat of antimicrobial resistance.

## Results

**hBD-3 analog is polymorphic in water and lipid bilayers.** To achieve site-specific resolution for NMR characterization, we synthesized the hBD-3 analog, a 45-residue peptide, using two isotope-labeling schemes (Fig. 1a). In total, nine $^{13}$C, $^{15}$N-labeled residues are included to cover both the N- and C-terminal halves of peptide sequence, as well as the different structural domains. Peptide 1 (VALIG) contains labels at V13, A19, L21, I30, and G37, while Peptide 2 (IVLG) has labeled residues of I3, V20, L24, and G31. All the cysteine residues are in the reduced form, which defines the analog state. Because hBD-3 is rich in cationic residues (13 Arg and Lys) that confer bactericidal activity[35], both peptides were reconstituted into POPC/POPG lipid bilayers that mimic the negatively charged state of bacterial membranes.

Figure 1b shows the representative 1D $^{13}$C spectra of hBD-3 analogs in POPC/POPG bilayers. With 65–70 wt% hydration, the hBD-3 analog exists in three major states with decreasing mobility: dissolved in solution, loosely associated with the membrane surface, and deeply inserted into the lipid bilayers. The highly mobile hBD-3 analogs dissolved in the aqueous phase is detected through the J-coupling-based refocused Insensitive Nuclei Enhanced by Polarization Transfer (INEPT) technique[36,37]. $^{13}$C direct polarization (DP) with a short recycle delay preferentially selects the relatively mobile components[38,39], with contributions from partially dissolved or loosely bound peptides, while $^{1}$H-$^{13}$C cross-polarization (CP) detects the peptides rigidified by their insertion in lipid bilayers. At 298 K, the well-inserted peptides are relatively rare as evidenced by the eight times stronger peptide signals in DP than in CP spectrum. The peptide peaks at 40–60 ppm region shows identical $^{13}$C chemical shifts in INEPT and DP spectrum, but change slightly from those in CP spectrum. Taken together, the hBD-3 analog undergoes minor structural changes upon binding to the membrane, whereas there is little difference between the dissolved state and the state with loose attachment to the membrane. As most peptides remain relatively mobile at high temperature and CP cannot provide sufficient sensitivity for two-dimensional (2D) experiments, a moderately lower temperature 269 K is used to overcome the sensitivity barrier for detecting membrane-bound peptides and to simultaneously retain the liquid-crystalline phase of the membrane as demonstrated by the sharp $^{1}$H peaks of lipid acyl chains (Supplementary Fig. 1)[40]. The 2D $^{13}$C–$^{13}$C correlation spectra provide atom-specific resolution of hBD-3 analog in both mobile and membrane-bound states (Fig. 1c; Supplementary Fig. 2), and all the $^{13}$C chemical shifts are documented in Supplementary Table 1.

The structure of hBD-3 analog is highly polymorphic when bound to phospholipid membranes. The dynamic peptides in water exhibit relatively sharp linewidths of 1–1.5 ppm for aliphatic carbons, because the conformational heterogeneity is averaged out by rapid motions (Fig. 1c). Most residues show a single set of chemical shifts, while residues I3, A19, V20, and I30 have an additional, minor set of peaks, indicating the presence of a less populated state in addition to the major conformer. In contrast, the rigid, membrane-bound peptides have substantially broader linewidths and more pronounced peak multiplicity. For example, three well-resolved Cα-Cβ cross peaks have been identified for I30 at (60, 37 ppm), (59, 41 ppm), and (57, 39 ppm). Similarly, we have identified three subtypes for I3, L21, and L24 residues, and two conformers for the other residues, indicating a highly polymorphic peptide structure induced by membrane interactions.

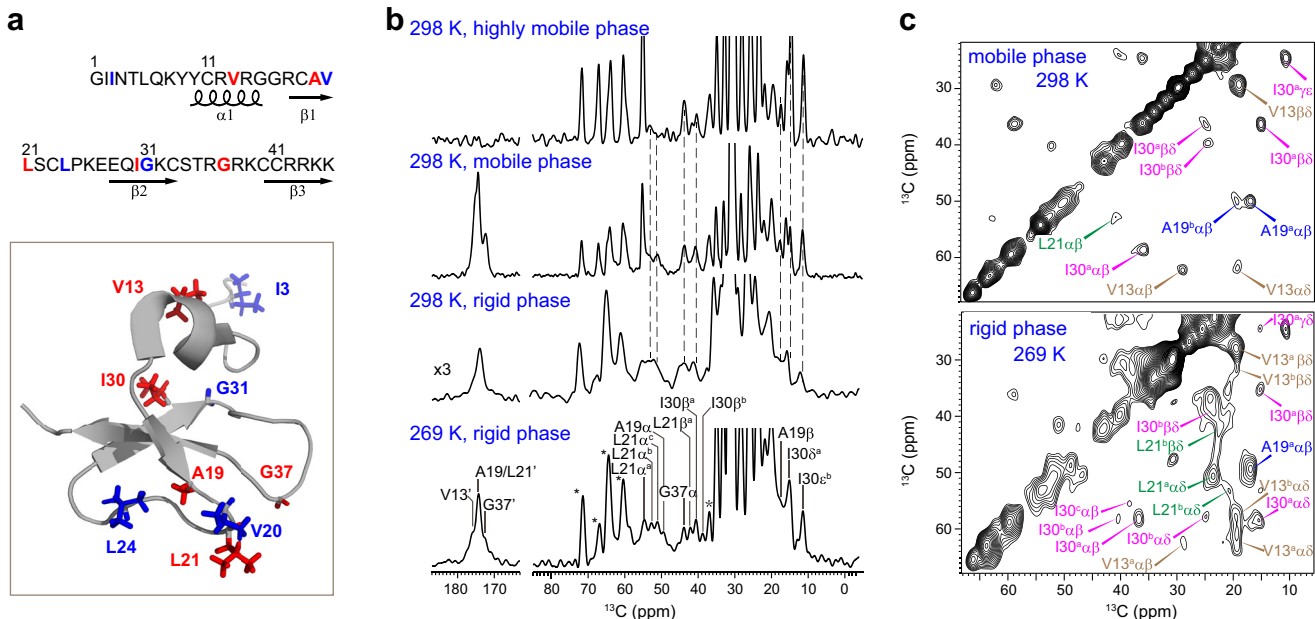

**Fig. 1** Isotope-labeling schemes and $^{13}C$ solid-state NMR spectra of hBD-3 analog. **a** The $^{13}C$, $^{15}N$-labeled residues are in red for VALIG peptide, and in blue for IVLG peptide in both the amino acid sequence and solution monomeric structure of wt-hBD-3 (PDB 1KJ6). **b** 1D $^{13}C$ spectra of VALIG peptide in POPC/POPG bilayers at 269 K and 298 K. From top to the bottom are INEPT, DP, and CP spectra measured at 298 K, and CP spectrum at 269 K. Dashlines indicate the resolved peptide peaks. **c** 2D $^{13}C$-$^{13}C$ correlation spectra of VALIG with 100-ms DARR mixing. The rigid and mobile components are selected using $^1H$–$^{13}C$ CP and $^{13}C$ DP, respectively. Greek letters indicate carbon sites and superscripts annotate conformers. I30$^a$αβ: carbon α to carbon β cross peak of subtype-a I30

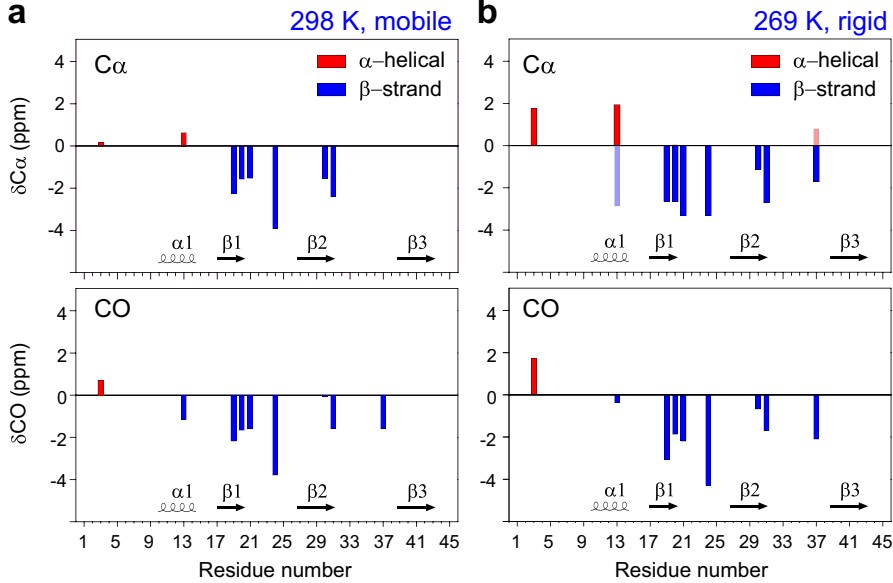

**Fig. 2** The hBD-3 analog mainly adopts β-strand conformation. $^{13}C$ secondary chemical shifts of Cα and CO of hBD-3 analog in POPC/POPG membranes at **a**, 298 K, DP and **b**, 269 K, CP. The secondary structures of the wt-hBD-3 are also shown in the bottom of each panel for comparison with the analog. Transparent bars indicate the minor conformers

**Dominant β-strand conformation in lipid bilayers**. The hBD-3 analogs are rich in β-strand conformation as revealed by the $^{13}C$ backbone chemical shifts that are sensitive to φ and ψ torsion angles (Fig. 2a)[41]. The chemical shift differences between the observed data and random coil values of Cα and CO are used to identify the secondary structure[42]. In the mobile phase, the major conformer of hBD-3 analog is predominantly in β-strand, except for the N-terminus, which is consistent with the solution-NMR

monomeric structure of hBD-3[18]. The secondary chemical shifts characterize I3 as α-helical conformation, but suggest V13 as part of the loop. Consistently, when bound to lipid bilayers, the overall β-strand conformation is retained (Fig. 2b). Compared with the aqueous phase, the membrane-bound peptides have a higher tendency for α-helical conformation at V13 position in its major conformer, but more pronounced β-strand conformation for the minor subtype. It should be noted that the presence of membrane

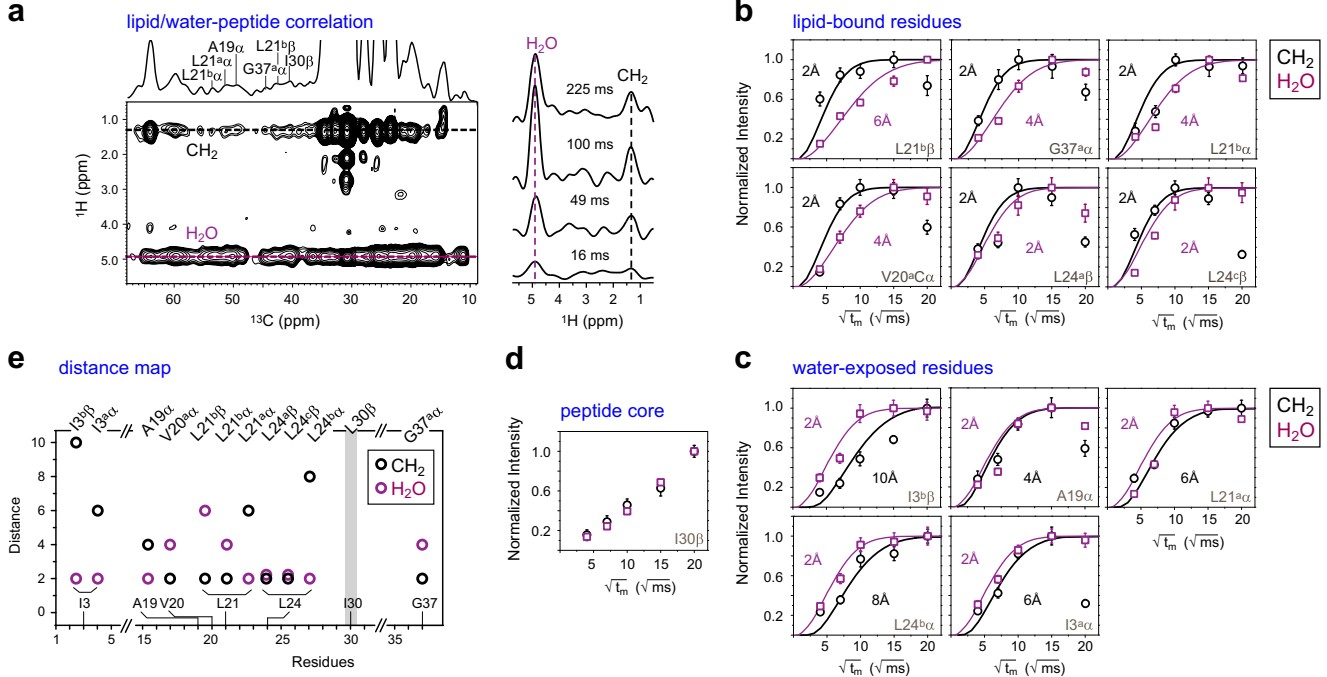

**Fig. 3** Site-specific depth of insertion of hBD-3 analog in POPC/POPG bilayers. **a** 2D $^{13}C$–$^1H$ correlation spectrum of VALIG, with 225-ms $^1H$ spin diffusion and the representative $^1H$ cross-sections of L21$^a$Cα at different $^1H$ mixing times. The water-to-peptide (purple) and lipid-to-peptide (black) $^1H$ polarization transfer curves of **b**, well-inserted residues and **c**, surface-bound residues are shown. The carbon site and the best-fit distances are labeled for each panel. Error bars are standard deviations that are propagated from signal-to-noise ratios. **d** I30 is far from both water and lipids. **e** Summary of distances from water and lipid acyl chains to the peptide

environment is crucial for the folding of hBD-3 analog as revealed by circular dichroism (CD) spectra (Supplementary Fig. 3).

**Insertion depth is site-specific and conformer-dependent.** Since hBDs inhibit bacteria by breaching their surface membrane, it is of high significance to determine the site-specific insertion topology of these peptides in bacteria-mimetic membranes. Herein, we conducted a series of 2D $^{13}C$-detected $^1H$ spin diffusion experiments that correlate the peptide $^{13}C$ signals with the lipid $CH_2$ (1.3 ppm) and water (4.9 ppm) $^1H$ signals, the intensities of which reflect the spatial proximity of each residue to the lipid acyl chain and surface water, respectively (Fig. 3a). With only a moderate $^1H$ mixing time of 100-225 ms, many peptide–lipid cross peaks already appeared, confirming the membrane insertion of hBD-3 analogs. In contrast, if a peptide only binds membrane surface, the paramyxovirus fusion peptide in DMPC bilayers for example, no peptide–lipid cross peaks are observed even at longer mixing times up to 900 ms[43,44].

The depth of insertion of hBD-3 analog is residue-specific, which is demonstrated by the wide range of semi-quantitative distances derived by fitting the intensity buildup curves using a 1D lattice model manipulating polarization transfer (Fig. 3b–d)[45]. The results are mainly categorized as membrane-inserted (Fig. 3b) and surface-bound residues (Fig. 3c). Residues G37$^a$, V20$^a$, L24$^a$, and L24$^c$ are found to be well-inserted, with the lipid $CH_2$ signals reaching plateau rapidly, within 100 ms, but much slower buildup curves for water intensities (Fig. 3b). Consequently, the lipid-to-peptide distance is best fit to 2 Å, whereas the distance from water to peptide varies from 2 to 6 Å. In contrast, L21$^a$, I3$^a$, I3$^b$, A19, and L24$^b$ are spatially proximal to water (2 Å), but far from lipid acyl chains (4–10 Å), thus residing on the membrane surface (Fig. 3c). It is unexpected that no equilibrium could be reached for I30 even with an extended mixing time of 400 ms, neither for water nor lipids (Fig. 3d). It suggests that I30 is embedded in the

hydrophobic core of the peptide, likely on a peptide–peptide interface, thus becoming inaccessible to external molecules.

The membrane-bound topology of hBD-3 analog is also found to be conformation-dependent. As revealed by the NMR-derived distance map (Fig. 3e), the polymorphic residue L21 has its major conformer (type-a, L21$^a$α) binding membrane surface and the minor conformer (type-b, L21$^b$ β) inserted in the hydrophobic core, whereas L24 has the reversed trend. The N-terminus of hBD-3 is consistently water-proximal, while the C-terminus is deeply inserted into the bilayers. These NMR data resolved the site- and conformer-specific membrane insertion of hBD analogs.

**Membrane-bound residues are rigidified.** To probe the peptide dynamics, we measured the motionally averaged $^{13}C$–$^1H$ dipolar coupling of each residue in POPC/G bilayers. With a dipolar dephasing period, the decline of peptide peak intensity indicates relatively strong $^{13}C$–$^1H$ dipolar couplings (Fig. 4a). Most of the dipolar evolution curves exhibit asymmetric patterns indicative of intermediate-time scale motion. Only I3 has a near-symmetric curve and a long $T_2$ relaxation time of 5 ms due to the high mobility and solvation of the N-terminus (Fig. 4b, c; Supplementary Table 2). The best-fit dipolar coupling constant and order parameters are listed for each carbon site.

The dynamics of membrane-bound hBD-3 analog is site-dependent. At 269 K, the Cα signals of I3, A19, L21$^b$, and G31 have small order parameters ranging from 0.50 to 0.64, indicating large-amplitude motions (Fig. 4b, d). On the contrary, larger order parameters of 0.73–0.82 have been observed for the backbone carbons of G37, V20, L24$^a$, and L24$^b$ (Fig. 4c, d): all these residues have shown deep insertion (Fig. 3e), which in turn confines their motion. At a higher temperature of 298 K, the order parameters are slightly lower for the membrane-inserted peptides (0.5–0.8) with V20 and L24 remaining as the most rigid

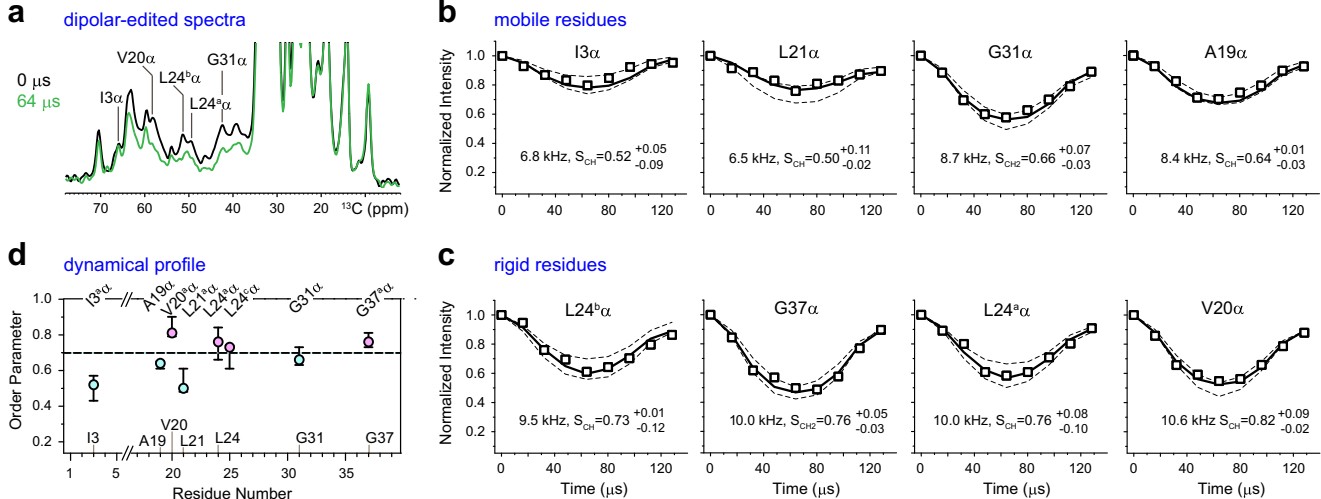

**Fig. 4** Mobility of hBD-3 analog in POPC/POPG bilayers at 269 K. **a** 1D $^{13}$C control, equilibrium spectrum (black) compared with a 64-µs dipolar-dephased spectrum (green), with signals from the rigid components dephased. $^{13}$C−$^{1}$H dipolar couplings curves are categorized as **b**, mobile and **c**, rigid residues. The best-fit dipolar couplings, the corresponding order parameters, and error bars are labeled. **d** C−H order parameters of membrane-bound peptides. The relatively rigid residues are highlighted in magenta and the mobile residues are in cyan. Error bars represent the well-fit order parameters obtained from panels **b** and **c**. Dashline indicates the order parameter of 0.7

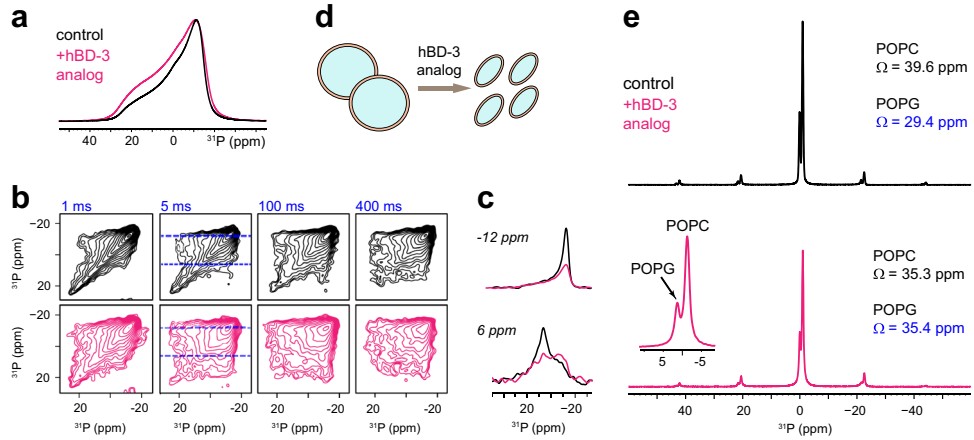

**Fig. 5** The hBD-3 analog perturbs membrane morphology and rigidifies negatively charged lipids. **a** $^{31}$P static spectra of control POPC/G membranes without (black) and with (magenta) hBD-3 analog at 298 K. **b** 2D static $^{31}$P–$^{31}$P exchange spectra measured with 1, 5, 100, and 400 ms mixing times. The hBD-bound sample (magenta) has more off-diagonal signals due to rapid exchange. **c** Cross-sections at 6 ppm and 12 ppm, with diagonal normalized (asterisk). **d** Illustration of the effect of hBD-3 analog on POPC/G vesicles. **e** 1D MAS $^{31}$P spectra shows resolved POPC and POPG signals. Fitting the sideband patterns provides information on the chemical shift anisotropy. Peptide-bound POPG has an increased span (Ω) due to reduced motions of headgroups

residues, but they decrease substantially to 0.3–0.5 for unbounded peptides (Supplementary Fig. 4, Supplementary Table 2).

Remarkably, the order parameters are consistent with the corresponding depths of insertion (Figs. 3e, 4d): only those residues on membrane surface can undergo large-amplitude motions, while those well-inserted residues are immobilized. Such high-resolution information is heretofore unavailable and will be integrated with MD modeling to detail the topology of hBD-3 analog in membranes as described later.

**Peptides disrupt membrane morphology and rigidify POPG lipid.** The dynamics, symmetry and phase, and surface curvature of phospholipid membranes can be closely monitored using $^{31}$P NMR. Adding peptides induces a considerable change in the static $^{31}$P spectral pattern (Fig. 5a), indicative of an altered distribution of lipid headgroup orientation: with hBD-3 analogs, the membrane should exist in ellipsoid shape instead of spheres, and

this NMR observation echoes the imaging results (Supplementary Fig. 6). The absence of a sharp, isotropic peak excludes the possibility of isotropic phases, such as micelles and cubic phase[46,47]. Hexagonal phase is not present either as the signature inverted powder pattern is missing[48,49].

The rate of phospholipid reorientation along the membrane surface due to the lateral diffusion is probed using 2D static $^{31}$P–$^{31}$P exchange spectra. Adding peptides increases the off-diagonal intensity (Fig. 5b, c; Supplementary Fig. 5), revealing a higher diffusion coefficient[50–52], and subsequently, a higher curvature and smaller vesicles. Therefore, hBD-3 analog has fragmented POPC/G membranes into smaller, ellipsoidal vesicles, which facilitates efficient reorientation of lipids (Fig. 5d).

Given the cationic state of hBDs, electrostatic interactions with negatively charged lipids may play a central role in stabilizing peptide–membrane interactions. Magic-angle spinning (MAS) unambiguously resolves the $^{31}$P signals of neutral POPC and

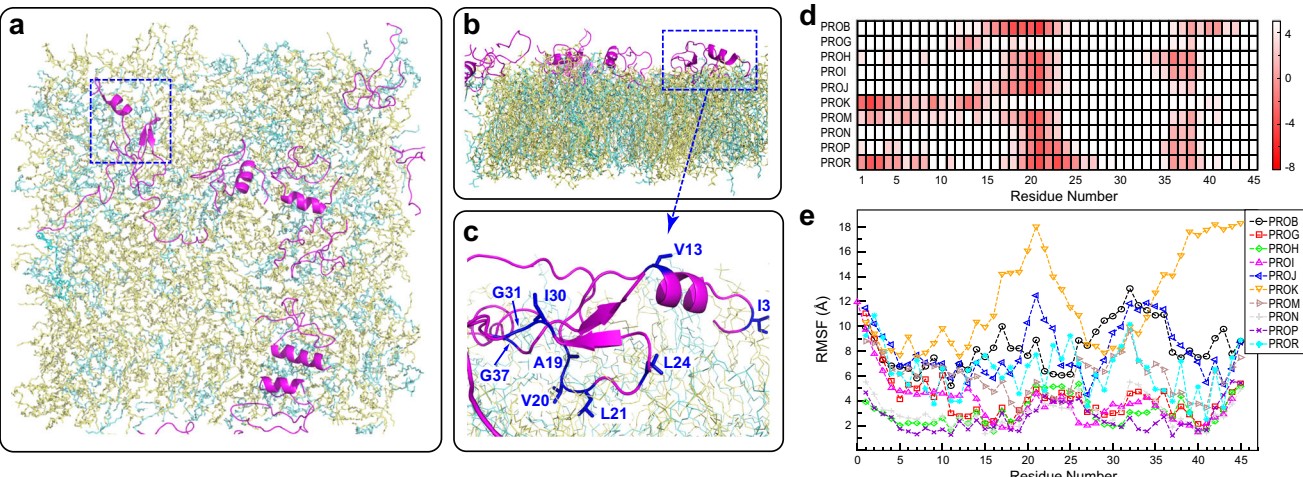

**Fig. 6** MD simulation reveals the insertion pattern of hBD-3 analog in POPC/G bilayers. **a** The topview and **b**, sideview of the last structure of hBD-3 analogs (magenta) in POPC/G from Anton Simulation. Water and ions are not shown. POPC and POPG lipids are in yellow and cyan, respectively. **c** Zoomed-in view of hBD-3 analog insertion through its loop regions. **d** Insertion depth matrix for ten peptides in POPC/G lipid bilayers. Residues 18–24 and 36–40 have deep insertion conserved in different peptides. **e** RMSF of hBD-3 analog binding POPC/G bilayers based on 5.0 μs Anton simulation trajectory

anionic POPG lipids (Fig. 5e) and fitting the spinning sideband intensities retrieves the [31]P chemical shift anisotropy (CSA) information reflective of lipids headgroup dynamics[53]. With peptides, the CSA span has decreased for POPC as expected for enhanced lipid mobility in more dynamic vesicles, but increased for POPG lipids (Fig. 5e; Supplementary Table 3). This is attributed to the tight association of POPG and peptide, which restricts the motion of phosphate headgroups in the negatively charged lipids.

**MD simulation reveals the inserted structure of hBD-3 analog.** To better understand the ssNMR experimental results and correlate them with peptide structure, we run a long-term (5.0 μs) Anton[54] MD simulation on the binding of 18 hBD-3 analogs to a bilayer formed by 480 POPC/POPG (3:1) lipids. After the peptides approach the membrane initially, during equilibration, 8 out of 18 peptides detached the membrane surface and partitioned into the aqueous phase, which explains the NMR sensitivity barrier for detecting membrane-bound peptides. The last structure of peptide–membrane complex (Fig. 6a, b) and a magnified view shows the tight membrane association for residues A19, V20, L21, and G37 (Fig. 6c). These residues are within two loops spanning from residue 18–24 and 36–40 in solution-NMR structure, which have deep insertion as revealed by the time- and lipid-averaged map of insertion depth (Fig. 6d). The matrix is obtained by calculating the average insertion depths of each residue of individual hBD-3 analog peptide into the top layer of membranes over the last 4.0-μs simulation (Supplementary Fig. 7). These simulation results dovetail with the [1]H spin diffusion results and dynamics measurements (Figs. 3, 4), providing a structural view that generally fulfils the NMR-derived restraints.

The root mean-squared fluctuation (RMSF) of the atomic positions in hBD-3 analog (including both peptide backbone and sidechains) reveals that only a few hBD units (PROP, PRON, and PROH) form stable binding with the lipid membranes, with much smaller RMSF values than the other peptides (Fig. 6e). The lipid-binding of PROG is also partially stable, but with larger-scale motions (presumably in solution) for the N-terminus, thus showing higher RMSF. In general, hBD-3 analogs exhibit the lowest RMSF for the regions of residue 17–22 and 36–39, which generally match the MD-derived distance map and ssNMR-restrained structural topology.

Notably, the analog partially inserts into the POPC/G mixture lipid bilayers during the self-assembly simulations (Supplementary Fig. 8a). In addition, clustering of POPG lipids to hBD-3 was observed because of the electrostatic interface between positively charged hBD-3 and negatively charged POPG lipids. In the self-assembly simulations of the peptide dimer with POPC/POPG lipids, there are more POPG lipids in the upper leaflet than in the lower leaflet, since the peptides stay on the upper leaflets. There are 21 POPG lipids in the upper layer while 15 POPG lipids in the lower leaflet based on trajectories in the last 50 ns MD simulations. Such observation echoes the preferential colocalization of hBD-3 analogs with POPG lipids (cyan) in the POPC/G mixture (Fig. 6a)[55] and the reduced mobility of these negatively charged lipids by peptide interactions (Fig. 5e). Therefore, the binding of hBD-3 promotes lipid clustering of POPG. Currently, there is no experimental evidence supporting the formation of lipid nanodomains[56], which needs follow-up investigations.

During the simulations of 18 peptides (nine pairs of dimers), all the dimers dissociated. That is also observed in the self-assembly simulations on hBD-3 dimer in analog forms in the POPC mixed with POPG lipids (Supplementary Fig. 8a). However, in a 300-ns MD simulation, it was found that hBD-3 can bind a bilayer containing only POPG lipids in a dimer form stably, with the binding interface consistent with ssNMR observations (Supplementary Fig. 8b).

## Discussion

To our knowledge, this study presents the first high-resolution investigation that integrates experimental ssNMR results with long-term MD modeling to reveal the function-relevant structure and dynamics of hBDs in lipid bilayers and its effects on membrane morphology. Three major findings are provided. First, despite the enhanced structural polymorphism, the secondary structural characteristics of hBD-3 analog are largely retained in both the solvated phase and membrane-bound state. Therefore, insignificant or local fluctuation of the peptide structure is sufficient for accommodating membrane insertion, which differs from the functional mechanism of many other proteins (e.g., fusion proteins) that often require conformational plasticity[49,57]. Second, the experimentally measured depth of insertion and membrane-confined dynamics dovetail with the MD-derived insertion depth matrix, collectively revealing a membrane-bound

topology of hBD-3, in which a structural surface that accommodates two adjacent loops is deeply embedded in lipid bilayers. Third, we find that hBDs specifically interact with and rigidify the negatively charged POPG, promotes lipid aggregation and separation, and further deform the membranes to produce smaller, non-spherical vesicles.

The prevailing, hypothetical models consider the positive charge (theoretical PI of 10.0), hydrophobic property, and the secondary structure of the peptide as the determinant of hBD's capability of disrupting phospholipid membranes[58–60]. Our results suggest a minor role of secondary structure, but demonstrate that the peptide adopts a specific orientation to insert into the membrane, with the region containing residues V20, L21, L24, and G37 being deeply inserted (Fig. 6c). These residues are spatially located at two adjacent loops of solution-NMR wt-hBD-3 structure, and are also highly conserved from mouse to chimpanzee (Supplementary Fig. 9), suggesting that these exposed hydrophobic residues and this specific surface binds membrane as a common feature in the β-defensin family. The potential interactions stabilizing this complex, e.g., hydrogen bonding between basic residues and membrane phosphate head group or electrostatic contacts, should be further restrained using $^{31}P–^{13}C$ or $^{2}H–^{13}C$ distances[61–65].

Although diffusion NMR, dynamic and static light scattering, and native gel-migration analysis have suggested that hBD-3 form a dimer in solution[18], the dimeric structure has not yet been determined. X-ray has revealed an intermolecular antiparallel β-sheet formed by the β1-strand of hBD-2 monomer[22], while solution-NMR of hBD-3 has proposed strand β-2 as another potential site for dimerization (Supplementary Fig. 10)[18,20]. Our $^{1}H$ spin diffusion data revealed that I30 is shielded in the hydrophobic core of peptide complex, with slow $^{1}H$ spin diffusion rate from both lipids and water. Based on the monomeric structure of wt-hBD-3, the solvent accessible surface area (SASA) calculated by PyMOL is moderate for I30 (30.4 Angstroms[2]) (Supplementary Table 4), thus this residue should be at least partially exposed in the monomer. Given the similar secondary structure between wt-hBD-3 and the analog, the observed inaccessibility of I30 is unexpected and should originate from the involvement of this residue in an oligomeric interface.

MD simulation provides a view of the potential dimer supporting the hypothesis of strand β2 as the dimer interface, with E28, Q29, and I30 involved in the dimer interface when binding on pure POPG lipid bilayer (Supplementary Fig. 8b). Our previous modeling study has suggested that, for hBD-3 analog with the disulfide bonds released in reducing condition, the dimer interfaces exhibit reduced stability in solution[66], it thus becomes important to understand how membrane interactions could support oligomerization. Also, the NMR results support the presence of oligomerized hBD-3 analogs in POPC/POPG bilayers, but the exact oligomeric number remains unclear and awaits experimental determination.

In conclusion, this study provides theretofore unavailable experimental, molecular-level evidence for understanding the functional structure and mechanism of β-defensin family. With corroborated observations from spectroscopic and modeling methods, we reveal that the peptide utilizes two highly conserved, adjacent loops in the solution-NMR structure for membrane partitioning. The secondary structural features and surface charges may help the peptide to gain the energetically favorable orientation to insert into and perturb the membrane. These novel findings provide the structural basis for optimizing and developing defensin mimetics against pathogenic infections, and more importantly, encourage many future investigations of the oligomerization state of defensins, their dependence on lipid

composition, and the functional structure and mechanism of many other relevant peptides and derivatives.

## Methods

**Peptide synthesis and purification.** The hBD-3 analog peptides of 45 amino acids were synthesized using Fmoc chemistry. Two isotope-labeling schemes were used to provide site-specific resolution and ensure coverage of the whole span of the peptide sequence. The VALIG sample contains $^{13}C$, $^{15}N$-labeled residues V13, A19, L21, I30, and G37, while IVLG sample contains $^{13}C$, $^{15}N$-labeled residues I3, V20, L24, and G31. $^{13}C,^{15}N$-labeled amino acids were protected by Fmoc in lab, with purity of > 90% as monitored by $^{1}H$ solution-NMR spectroscopy. The peptides were synthesized using Fmoc solid-phase methods using 0.1 mM scale with five-fold excess input for the labeled residues to ensure efficiency in the double couplings. The Fmoc group was then removed with 20% piperidine twice (3 min and 10 min) at room temperature, and the resin was washed with DMF and DCM. The peptide was cleaved from the resin and sidechain deprotected using TFA/EDT/water (4 mL, 94:3:3) for 5 hr. The cleavage reaction was repeated for 10 min. The crude peptide was then purified by HPLC. Eluents for A is 0.1% TFA in water and that for B is 0.1% TFA in acetonitrile. MALDI–TOF analysis was performed to verify the molecular weight of peptides. In total, 5184.8 Da for VALIG and 5181.0 Da for IVLG are the match to the calculated molecular weight. The purity is higher than 98% for both peptides.

**Membrane-bound peptide sample preparation.** Lipids 1-palmitoyl-2-oleoyl-sn-glycero-3-phosphocholine (POPC) and 1-palmitoyl-2-oleoyl-sn-glycero-3-phosphatidylglycerol (POPG) at a molar ratio of 3:1 were dissolved in chloroform, and dried using nitrogen gas to form a mixed lipid film. The residual solvents were completely removed under vacuum overnight. The lipid mixture powder was resuspended in 10 mM HEPES buffer (pH 7.0). To prepare unilamellar vesicles, the lipid mixture undergoes six cycles of freeze-thawing using 40 °C water bath and liquid nitrogen. Simultaneously, the peptide solution is prepared by adding 400 μL of HEPES buffer (10 mM, pH 7.0) to ~10 mg of the synthesized peptide. The liposome solution and peptide solution were mixed, which results in precipitation immediately, and this method has been used previously for preparing the hBD-3 analog-containing membranes for $^{2}H$ ssNMR studies[67]. We chose a condensed concentration of peptide, with a peptide-to-lipid molar ratio of 1:14, which converts to a mass ratio of 1:2 to improve the population of peptides partitioning into the membrane phase and ensure sufficient NMR sensitivity. Since most peptides remain dissolved and only a minor portion stays in the membrane, the actual peptide-to-lipid ratio for the membrane-bound state should be well below 1:100. The sample was spun down by centrifugation at 10000×g for 20 min. The supernatant was removed. The pellet was incubated in the desiccator overnight to slowly remove the excess water, reduce the hydration to 65–70%, and packed into 4-mm MAS rotors with a Kel-F insert.

**Solid-state NMR experiments.** All the solid-state NMR experiments were performed on a Bruker 400 MHz (9.4 Tesla) spectrometer. The radiofrequency field strengths are 71 kHz for $^{1}H$ decoupling and 50–62.5 kHz for $^{13}C$. The $^{13}C$ chemical shifts were externally referenced to the TMS scale by calibrating the Met Cδ peak of model peptide N-formyl-Met-Leu-Phe-OH (f-MLF)[68] at 14.0 ppm. Most of the spectra were collected under 10 kHz magic-angle spinning (MAS) at 298 K or 269 K. The typical recycle delays were 1.5–2.0 s.

To measure the $^{13}C$ chemical shifts of hBD-3 analog, we measured a series of 2D $^{13}C–^{13}C$ Dipolar-Assisted Rotational Resonance (DARR) spectra with a moderate mixing time of 100 ms[69]. At 298 K, DP using a 90° $^{13}C$ pulse with a short recycle delay of 2 s was used to detect the mobile hBD-3 analogs in aqueous phase, and CP was used to select the rigid, membrane-bound state. A moderately low temperature of 269 K was chosen to partially immobilize peptides and enhance CP sensitivity. This temperature was measured by a thermocouple that was a few millimeters from the NMR rotor. Due to the heating effect of MAS, the real sample temperature is estimated to be 2–5 °C higher. Because of the structural polymorph, the coexistence of water and membrane phases, and the high solubility of hBD peptides, the sensitivity becomes extremely challenging for CP-based experiments. Despite the large amount of peptide (8 mg per sample), it still requires weeks of NMR time (1920 scans) for measuring each 2D spectrum.

To determine the insertion depth, we measured 2D $^{13}C$-detected $^{1}H$ spin diffusion spectra at 269 K. By using a $^{1}H$ $T_2$ filter of 2 × 0.5 ms, the $^{1}H$ magnetization from mobile lipids and water was selected and then transferred to peptides through a mixing period, and further to the $^{13}C$ nuclei for site-specific detection[45,70]. The semi-quantitative distances from peptide to water or lipid chains are obtained by fitting the buildup curves of peptide peak intensities as a function of the square root of mixing times. All buildup curves were corrected by $T_1$ relaxation. The diffusion coefficients for the lipid and peptide were $D_L$ of 0.012 $nm^2/ms$, $D_W$ of 0.03 $nm^2/ms$, respectively; the diffusion coefficients for the sink peptide was $D_P$ of 0.3 $nm^2/ms$. For the lipid–peptide interface, $D_I$ as 0.00125 $nm^2/ms$ for $H_2O$ and 0.0025 $nm^2/ms$ for $CH_2$ were used. These values have been widely used for antimicrobial peptides, DNA, and membrane channels[44,45,71].

To probe the peptide dynamics, we conducted $^{13}C–^{1}H$ dipolar-chemical-shift (DIPSHIFT) experiment[72] at 269 K under 7.5 kHz MAS. Frequency-Switched

Lee-Goldburg (FSLG) scheme[73] was used for $^1H$ homonuclear decoupling: the transverse $^1H$ field strength is 83.3 kHz, converting to effective field strengths of 102 kHz. The $^{13}C$ π pulse is using a field strength of 62.5 kHz. The number of t2 points is nine with an increment of 16.03 μs. Dipolar curves obtained by plotting the peak intensity as a function of dipolar evolution time were then fit using a Fortran program to obtain the apparent CH or $CH_2$ dipolar couplings. The rigid-limit dipolar coupling value is 22.7 kHz, which leads to the FSLG-scaled, rigid-limit coupling value of 13.1 kHz (scaling factor 0.577). Order parameters are calculated as the ratio between the measured couplings and the apparent rigid-limit value.

To examine the membrane morphology, we collected 1D and 2D $^{31}P$ spectra for POPC/POPG control sample and hBD-3-containing sample at 298 K. The $^{31}P$ chemical shift was referenced on the phosphoric acid scale. The control sample was prepared as described above, but with two cycles of freeze-thawing using liquid nitrogen and room-temperature water bath. In all $^{31}P$ experiments, the radiofrequency field strength of 50 kHz for $^{31}P$ was used with recycle delays of 1.5 s. Two-pulse phase-modulated (TPPM) decoupling sequence with $^1H$ decoupling field strength of 50 kHz was used in both 1D static and MAS $^{31}P$ spectra. Under static condition, 1D $^{31}P$ DP spectra were measured to probe lipid orientation and membrane morphology. 2D static $^{31}P$–$^{31}P$ exchange spectra with mixing times of 1, 5, 100, and 400 ms were measured to probe lipid lateral diffusion[50,51]. 2D experiments were initiated from direct $^{31}P$ polarization, and no $^1H$ decoupling was used during the mixing period. 1D $^{31}P$ DP and CP spectra were also measured under slow MAS of 3.5 kHz to resolve the POPC and POPG signals, and derive the $^{31}P$ CSA parameters based on the spinning sideband intensities using the Herzfeld–Berger analysis program[53].

**Circular dichroism measurement**. To measure the circular dichroism (CD) spectrum, both the solution state sample and membrane-bound state sample were prepared. For the solution state sample, peptides were dissolved in HEPES buffer (10 mM, pH 7.0) to reach a final concentration of 0.3 mg/mL (57 μM). For the membrane-bound state sample, POPC and POPG lipids at a molar ratio of 3:1 were mixed, dried, and resuspended in the HEPES buffer (10 mM, pH 7.0) as described above. The lipid mixture was treated with two freeze-thaw cycles in liquid nitrogen and room-temperature water bath. The peptide sample was then added into the liposome solution. The final concentrations of total lipids and peptide are 1.74 mg/mL and 0.3 mg/mL (57 μM), respectively. A lipid-only control sample was prepared with the same protocol without adding the peptide.

CD spectra were measured at 293 K on a Jasco J-815 CD spectrometer using a 1-mm quartz cuvette. Each spectrum had three replicated scans. It was processed through baseline correction, control sample subtraction, and smoothing sequentially. The smoothing was carried out using the Savitsky-Golay smoothing algorithm with the order of nine, The deconvolution of CD spectra was conducted using the BestSel web server[74].

**Transmission electron microscope measurement**. The same peptide-containing and peptide-free samples were prepared as described for the CD measurement. The liposome solution was diluted to 0.03 mg/mL before Transmission Electron Microscope (TEM) measurements. In total, 3 μL of each sample was placed onto a glow discharged TEM grid for several minutes, and then was negatively stained using 2% uranyl acetate solution. A very thin film was spanned on the grid by removing the excess solution with the paper filter. The TEM images were collected on the JEOL JEM-1400 electron microscope.

**MD modeling of hBD-3 analog in lipid bilayers**. Long-term Anton simulations on 9 hBD-3 dimers in the analog form disrupting POPC/G lipid bilayer were conducted to understand the peptide–lipid interactions that are probed experimentally using ssNMR. The dimer structure was predicted previously[66], and the peptide has all three disulfide bonds disconnected, which should break in a specific pathway in the reduced condition[66]. We set up all-atom CHARMM molecular dynamics simulations by placing nine hBD-3 dimers in analog form (18 hBD-3 units from PROA to PROR) above a bilayer of 480 POPC/POPG (3:1) lipids by at least 8 Å using the CHARMM-GUI online program[75–77] and CHARMM36m forcefield[78]. Therefore, the top and bottom layer each contains 180 POPC and 60 POPG lipids. TIP3P water molecules were added to solvate the system with at least 12 Å of water above the top and below the bottom of the peptides/lipids. Counter ions were added to neutralize the system in addition of 0.15 M of NaCl at 300 K and 1 atm. For nonbonded calculations, a cutoff of 12 Å was used. All bonds involving hydrogens were kept rigid using the SHAKE algorithm. After a brief energy minimization, the 20-ns equilibration run using NAMD program[79] with a time step of 2 fs, the simulation was continued on the Anton 2 supercomputer[54] for 5.0 μs. An NPT ensemble was applied on Anton simulation with a time step of 2.5 fs and a trajectory output frequency of 240 ps.

Using the above method, we also set up the hBD-3 dimer in analog form binding on a pure POPG bilayer that contains 72 POPG lipids on each layer. After setting up the system using CHARMM-GUI program, NAMD all-atom molecular dynamics simulation was conducted for 300 ns at 300 K and 1 atm. The final binding structure of hBD-3 dimer on POPG is shown in Supplementary Fig. 8b.

To determine the favorable binding location of hBD-3 analog with membranes, we set up a self-assembly simulation of hBD-3 analog with POPC/G mixture. The

simulation contains a dimer mixed with randomly packed 108 POPC and 36 POPG lipids, 8425 TIP3P water molecules, and 42 SOD and 28 CLA ions spaciously (Supplementary Fig. 8a). The initial box size is 211 × 212 × 256 Å, which allows molecular rearrangement and assembly. After a brief energy minimization, the simulation was conducted on an NPT ensemble using all-atom NAMD simulations for 600 ns, the final box is 103 × 61 × 76 Å.

**Statistics and reproducibility**. Two $^{13}C,^{15}N$-labeled peptides and an unlabeled peptide were measured. All attempts for replication were successful. All two-dimensional experiments were carried out for 3–10 times and added together.

**Reporting summary**. Further information on research design is available in the Nature Research Reporting Summary linked to this article.

## Data availability
The data that support the findings of this study are available from the corresponding authors upon request. The data include solid-state NMR data (Bruker Topspin files) and MD modeling files, which are currently stored in computers. The Source Data underlying Figs. 2a, 2b, 3b–d, 4b, 4c, 6d, and 6e, as well as Supplementary Figs 4c, 4d, and 7 are provided as Supplementary Data 1.

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

## Acknowledgements

The project was supported by the Burroughs Wellcome Fund (BWF) 2018 Collaborative Research Travel Grant (CRTG) to L.Z. T.W. thanks funding support from National Science Foundation under the award number of NSF OIA-1833040. Anton 2 computer time was provided by the Pittsburgh Supercomputing Center (PSC) through Grant R01GM116961 from the National Institutes of Health. The Anton 2 machine at PSC was generously made available by D.E. Shaw Research. Some of the short-term simulations were performed on the high-performance computer resource on TTU campus and on Bridges located in Pittsburg Supercomputer Center. The authors would like to thank Ms. Ying Xiao and Dr. Ted Gauthier for experimental assistance. TEM measurements were performed at the Shared Instrumentation Facility (SIF) at Louisiana State University.

## Author contributions

X.K., A.K., and T.W. conducted the NMR, CD, and TEM experiments and analyzed the experimental data. C.E., J.P., and L.Z. conducted the MD simulations and analyzed the modeling data. A.K. and A.C. prepared the peptide and NMR samples. X.K., L.Z., and T.W. wrote the paper.

## Competing interests

The authors declare no competing interests.
