## [Peer Review File · Communications Biology]

Reviewers' comments:

Reviewer #1 (Remarks to the Author):

This manuscript presents a study on the interaction of reduced hbd-3 analog with bilayers mimicking bacterial membrane using a combination of NMR and molecular dynamics simulations. The reduced hbd-3 analogue shows enhanced conformational diversity than the wildtype peptide, both in solution and on membrane. Using ssNMR, the authors identified two loops, mainly hydrophobic residues, to insert deeper into the membrane, which was consistent with the results of MD simulations. The NMR results also suggested oligomerization of the peptide and preferential interactions with the negatively charged POPG lipids. Overall this study is well performed and the results are detailed and discussed. The findings could be useful to readers on peptide-membrane systems. However, there are some issues to be addressed to improve the manuscript.

1. The reduced form of hbd-3 is expected to have higher conformational flexibility, and the authors have discussed the secondary structure in the NMR results, it would be useful to include additional experiments (e.g., circular dichroism experiment) to compare the secondary structure of reduced hbd-3 with wt-hbd-3 both in solution and in the lipid environment.
2. Suppl. Fig. 9 shows two dimeric structures of wt-hbd-3 derived from x-ray and NMR, which are quite different. It is not clear what the conformation of dimer structure is used in the MD simulation. The authors should include their dimeric structure. Moreover, during the MD simulations of 18 peptides with membrane, are the dimers stable?
3. In the self-assembly simulations of peptide dimer with POPC/POPG lipids, since the peptides stay on the upper leaflet, are there more POPG lipids in the upper leaflet than in the lower leaflet?

Minor issues:

4. The peptide used is the reduced form of hbd-3, but the authors used different terminologies, such as hbd-3 analog, hbd-3, hbd, etc. I suggest to use a consistent name for the peptide.
5. In the methods of Peptide synthesis, it is mentioned that "518 Da for IVLG", is it a typo?
6. In the methods of MD simulations: "the top and bottom layer each contain 160 POPC and 80 POPG lipids". The ratio is 2:1, not 3:1.

Reviewer #2 (Remarks to the Author):

In this manuscript, Xue et al. describe their study on how an analog of human defensin interacts with the lipid bilayer, which is important to understand the antimicrobial function of this peptide. Even though the peptide is polymorphic, the authors find that it does not need big conformational change to interact with the lipid. The authors then apply the spin diffusion technique to categorize the residues into groups close to water or close to lipid, which provide site-specific information on the residues inserted into the lipid. Through the order parameter measurement, the authors show that residues get more rigid after interacting with lipid. On the other hand, the lipid shows more lateral diffusion due to the peptide interaction, which the authors assume is caused by changed curvature of the lipid. However, the negatively charged head group PG gets more rigid due to the electrostatic interaction between the peptide and lipid based on the Chemical Shift Anisotropy analysis. This is more clearly seen in the MD simulation where POPG is clustered. Overall, this is an excellent study showing the power of combining SSNMR and MD to study a difficult and important question. The experiments are strategically planned and the data is sufficient to support their claims for most of the time. I think it will appeal to a broad audience and will be valuable for the biophysical society.

Reviewer #3 (Remarks to the Author):

The manuscript by Kang et al report on site-specific structure and dynamics of human β -defensins in lipid bilayers by solid-state NMR experiments and molecular dynamic simulation. This study provides atomic level structural information crucial for understanding structure-function relationship of β -defensin and its effect in membrane-disruption. The β -defensin is highly polymorphic and retained the beta sheet structure when bound to membranes based on the ^{13}C - ^{13}C NMR experiments. Furthermore, the residue-specific insertion depths of β -defensin were determined by fitting the build-up curves of ^1H - ^{13}C correlated signals, and conformational mobilities of each residue were identified by dipolar coupling measurements. The phosphorus chemical shift anisotropy measurements indicate that the lipid phosphate headgroups were rigidified and the reorientation rates of lipids were extracted from 2D ^{31}P - ^{31}P spectra with different mixing time. The results and conclusions of different NMR experiments are quite consistent, and in combination with MD simulation the binding mode of β -defensin peptide to lipid bilayers is revealed. The investigation strategy is impressive, with the NMR data well interpreted and clearly presented, and the conclusions well validated. More importantly, the investigators employed and present a systematic approach to investigate protein dynamics and protein-lipid/protein-protein interactions, and the integrated solid-state NMR method could be applied to a wide range of biological systems. There are multiple novelties in the experimental design, including the idea of applying specific isotope-labeling schemes and 2D ^{31}P - ^{31}P exchange solid-state NMR experiments. Overall the manuscript is excellent and well written, and will be important to both biology community and solid-state NMR community.

There are a few comments that may require minor revisions and more explanations from the authors.

^{13}C spectra of mobile and rigid phase

While I highly agree with the site-specific data analysis of the 2D C-C spectra at different temperatures, I wonder how one can demonstrate that the mobile phase and rigid phase corresponds to dissolved and membrane-bound states, respectively. An explanation for the relationship between these concepts will help readers to clearly understand the data interpretation as the mobile/rigid phases and dissolved/membrane-bound states are frequently mentioned. For this biological system, the changes in spectral resolution and sensitivity may arise from several aspects: i) overall temperature dependent dynamics; ii) changes of insertion depth and membrane-bound state, which affects various level of dynamics changes in different segments. Consequently, is it possible that at 298K the molecules are mobile and homogeneous but are still bound to membranes instead of being in dissolved state?

Methods: solid-state NMR experiments

The experiments were well described, but some important parameters were absent for multiple NMR experiments (including some listed below but not limited to). Please also indicate whether the ^1H decoupling was applied during the ^{31}P experiments. Although the paper will be presented to general audiences, these experimental parameters are important for researchers to reproduce the same type of NMR experiments.

Conclusions

At the end of discussion, it will be helpful to remind readers the key findings by reorganizing and integrating the conclusions of each section.

Minor:

Page 3, line 50: "The solution or crystallographic structure of five β -defensins..." should be "...solution

or crystallographic structures...”

Page 4, line 67: “Here we integrate 2D ¹³C-¹³C, ¹³C-¹H, and ³¹P-³¹P correlation solid-state NMR...” should be “...³¹P-³¹P correlated...”

Page 6, line 101: the values of NMR experimental parameters should be described in detail for both 1D and 2D ¹³C experiments either in line or in experimental section. In particular, the duration of recycle delay and whether it is 90 degree or small angle excitation for ¹³C DP experiment.

Page 7, line 133: “...suggesting this region has high conformational flexibility...” the observations more likely suggest the existence of conformers or conformational heterogeneity. It is not very convincing that these regions are highly flexible based on the changes in secondary structure.

Page 14, line 280: “...reveal the function-relevant structure of hBDs in lipid bilayers and its effects on membrane morphology.” The significance of this work also includes the function-relevant dynamics. The statement can be revised as “the function-relevant structure and dynamics” throughout the manuscript.

Page 20, line 400: It is recommended to indicate the parameters such as recycle delay and RF field strength of ³¹P pulses regarding to ³¹P NMR experiments.

Reviewer #4 (Remarks to the Author):

Kang et al present a nice study of the structure, dynamics, and interaction of defensin with membranes by ssNMR and MD, which improves our understanding of the functionality of this antimicrobial peptide. The authors discussed the morphology, polymorphism, and dynamics of the peptide in both “dissolved” and rigidified forms. More importantly, they determined the topology and binding interfaces of the peptide with the membrane combining ssNMR and MD observations. They also shed light on how the peptide modulates the membrane structure. Their results provide great insights into the interaction of defensin and membrane with high-resolution.

Major questions:

1. Have you done J coupling-based experiments to distinguish the mobile phase? In DP spectra both mobile and rigid components show up. DP being more sensitive than CP suggests the presence of mobile components whereas DARR is not the experiment for their detection. Signals shown in DARR suggest somewhat rigidity for dipolar coupling signal transfer, therefore residues shown in Fig. 1c (298 K) can't be dissolved to retain sufficient dipolar coupling transfer. J coupling-based experiments need to be done for the detection of mobile components.
2. Can the authors perform further experiments to determine the oligomeric state of the membrane-bound peptide?
3. Does the polymorphism of the peptide have anything to do with its oligomeric state?
4. Can the authors provide other measurements to show the perturbation of membrane morphology, such as DLS? It is necessary to show hBD is functional active under experimental conditions.

Minor questions:

1. To quantitatively evaluate the dissolved and bound peptides, do you compare peak height or volume in DP and CP spectra? I wonder why there are dissolved peptide left in the pellet after centrifugation.
2. If the peptide is dissolved in solution, will you expect to see peaks in a DARR experiment? Could the dissolved form loosely tethered?

3. With longer spin diffusion times, do you see the decrease of diagonal peak intensity as with increases of off-diagonal peak intensities? As both diagonal and off-diagonal peaks change their intensity in the spin diffusion process, it makes sense to normalize according to sample amount rather than diagonal peak height. Also is there a particular reason for showing cross sections at 6 and -12 ppm?
4. What is the recoupling pulse power used in DIPSHIFT experiment? What is the number of t_2 points and its increment?
5. Please correct typos such as I30->G31 and 518 Da (?) for IVLG on Page 16.

Response to Reviewers' comments:

Reviewer #1

This manuscript presents a study on the interaction of reduced hbd-3 analog with bilayers mimicking bacterial membrane using a combination of NMR and molecular dynamics simulations. The reduced hbd-3 analogue shows enhanced conformational diversity than the wildtype peptide, both in solution and on membrane. Using ssNMR, the authors identified two loops, mainly hydrophobic residues, to insert deeper into the membrane, which was consistent with the results of MD simulations. The NMR results also suggested oligomerization of the peptide and preferential interactions with the negatively charged POPG lipids. Overall this study is well performed and the results are detailed discussed. The findings could be useful to readers on peptide-membrane systems. However, there are some issues to be addressed to improve the manuscript.

We thank the reviewer for acknowledging the soundness of the study.

1. The reduced form of hbd-3 is expected to have higher conformational flexibility, and the authors have discussed the secondary structure in the NMR results, it would be useful to include additional experiments (e.g., circular dichroism experiment) to compare the secondary structure of reduced hbd-3 with wt-hbd-3 both in solution and in the lipid environment.

We thank reviewer for the suggestion. Circular dichroism (CD) is used extensively for rapidly evaluating the secondary structure of macromolecules. We measured the CD spectra of hBD-3 analog in both the solution state and membrane-bound state as shown below. The spectrum of solution state of hBD-3 analog shows minimal helix component, whereas the spectrum of membrane-bound state of hBD-3 reveals substantial amount of helix feature, suggesting large conformational change between the fully dissolved and dispersed state, and the states when membranes are present.

As CD spectra show the composite results of all the coexisting states in the sample but are lacking specificity to resolve the individual state, conformer, or carbon sites that were revealed using NMR spectroscopy. We have now included this figure as Supplementary Fig. 3. and added a brief explanation in the figure captions.

2. Suppl. Fig. 9 shows two dimeric structures of wt-hbd-3 derived from x-ray and NMR, which are quite different. It is not clear what the conformation of dimer structure is used in the MD simulation. The authors should include their dimeric structure. Moreover, during the MD simulations of 18 peptides with membrane, are the dimers stable?

The original dimer structure of hBD-3 was predicted using all-atom Molecular dynamics simulations by Zhang group as in ref [65] in the maintext. The dimer structure predicted can agree with solution NMR structure obtained by Schibli et al, which is shown in Figure S10(b) in Supplemental Material. This is the original dimer structure used in the MD set up. We have now clarified it in the captions of Supplementary Fig. 10.

The dimers dissociate during MD simulation. We have now added a clarification: "During the simulations of 18 peptides (9 pairs of dimers), all the dimers dissociated. That is also observed in the self-assembly simulations on hBD-3 dimer in analog forms in the POPC mixed with POPG lipids (Supplemental Fig 8a). However, in a 300-ns MD simulation, it was found that hBD-3 can bind a bilayer containing only POPG lipids in a dimer form stably....."

3. In the self-assembly simulations of peptide dimer with POPC/POPG lipids, since the peptides stay on the upper leaflet, are there more POPG lipids in the upper leaflet than in the lower leaflet?

In the self-assembly simulations of peptide dimer with POPC/POPG lipids, since the peptides stay on the upper leaflets, there are 21 POPG lipids in the upper layer while 15 POPG lipids in the lower leaflet based on trajectories in the last 50 ns MD simulations. So, there are more POPG lipids in the upper leaflet than in the lower leaflet. We have now explained this point in the MD part of Result sections: "In addition, clustering of POPG lipids to hBD-3 was observed because of the electrostatic interface between positively charged hBD-3 and negatively charged POPG lipids. In the self-assembly simulations of the peptide dimer with POPC/POPG lipids, there are more POPG lipids in the upper leaflet than in the lower leaflet since the peptides stay on the upper leaflets. There are 21 POPG lipids in the upper layer while 15 POPG lipids in the lower leaflet based on trajectories in the last 50 ns MD simulations."

Minor issues:

4. The peptide used is the reduced form of hbd-3, but the authors used different terminologies, such as hbd-3 analog, hbd-3, hbd, etc. I suggest to use a consistent name for the peptide.

We apologized for the confusion caused by the inconsistency. We have now updated all the names as following: hBD-3 analog is the reduced peptide synthesized in this study, wt-hbd-3 is the wide-type hBD-3 processing 3 disulfide-bonds in solution state, and hBD(s) are the whole family of mammalian beta-type antimicrobial peptides.

5. In the methods of Peptide synthesis, it is mentioned that "518 Da for IVLG", is it a typo?

We thank the reviewer for pointing out this typo. We have corrected it to 5181.0 Da.

6. In the methods of MD simulations: "the top and bottom layer each contain 160 POPC and 80 POPG lipids". The ratio is 2:1, not 3:1.

Thanks for pointing it out. There are 180 POPC and 60 POPG lipids in each layer and the ratio of POPC to POPG is 3:1. We have now corrected the mistake.

Reviewer #2

In this manuscript, Xue et al. describe their study on how an analog of human defensin interacts with the lipid bilayer, which is important to understand the antimicrobial function of this peptide. Even though the peptide is polymorphic, the authors find that it does not need big conformational change to interact with the lipid. The authors then apply the spin diffusion technique to categorize the residues into groups close to water or close to lipid, which provide site-specific information on the residues inserted into the lipid. Through the order parameter measurement, the authors show that residues get more rigid after interacting with lipid. On the other hand, the lipid shows more lateral diffusion due to the peptide interaction, which the authors assume is caused by changed curvature of the lipid. However, the negatively charged head group PG gets more rigid due to the electrostatic interaction between the peptide and lipid based on the Chemical Shift Anisotropy analysis.

This is more clearly seen in the MD simulation where POPG is clustered. Overall, this is an excellent study showing the power of combining SSNMR and MD to study a difficult and important question. The experiments are strategically planned and the data is sufficient to support their claims for most of the time. I think it will appeal to a broad audience and will be valuable for the biophysical society.

We thank the reviewer for positive comments about the completeness and significance of this work.

However, I do have a few questions and suggestions, which are listed below:

1) In this study, the reduced form of the analog was used. How much is the difference between the reduced form and the un-reduced form? Is the crystal structure shown in the figure (Figure 1a) and in the MD simulation responding to the reduced form or the un-reduced?

Fig. 1a shows the structure of the un-reduced form. We have now added the name of wt-hBD-3 in the figure captions. The reduced form is the analog form, while the un-reduced form is the wildtype form. The MD simulation uses the reduced form, which is, the analog.

To date, there is not enough experimental data directly comparing the structure of wt-hBD-3 and the analog, but Fig. 2 provides the first view that the secondary structure is largely preserved in the analog sample and the solution-structure of wt-hBD-3.

2) At the line 101, could you provide more information on why the DP with a short cycle delay and CP can be used to probe the peptide in the aqueous solution and peptide on the lipid? Normally, an INEPT sequence is used to probe the residues with high mobility. Could you comment on why a short pulse delay was chosen?

We thank the reviewer for the advice. We have now added INEPT in Fig.1b and substantially expanded the second paragraph of the Results section to better clarify the

peptide states as well as the different polarization techniques (cited below). The ^{13}C DP with short recycle delays are selecting the mobile components with short ^{13}C T_1 relaxations, and the rigid components require much longer recycle delays before fully relaxing back to equilibrium. This method has been used in selecting partially dissolved peptide/protein domains (reference 38 and 39) or other molecules such as carbohydrates.

“With 65-70 wt% hydration, the hBD-3 analog exists in three major states with decreasing mobility: dissolved in solution, loosely associated with the membrane surface, and deeply inserted into the lipid bilayers. The highly mobile hBD-3 analogs dissolved in the aqueous phase is detected through the J-coupling-based refocused Insensitive Nuclei Enhanced by Polarization Transfer (INEPT) technique^{36, 37}. ^{13}C direct polarization (DP) with a short recycle delay preferentially selects the relatively mobile components^{38, 39}, with contributions from dissolved or loosely bound peptides, while ^1H - ^{13}C cross-polarization (CP) detects the peptides rigidified by their insertion in lipid bilayers.”

3) At the line 104, the authors made a claim that more peptide is in the aqueous solution because of the observation that the DP signal is 8 times stronger than the CP signal. Have the authors considered the possibility that the higher temperature could cause more mobility in the peptide bound to the lipid, thus cause a less efficient CP transfer?

We agree with the reviewer that high temperature could substantially reduce the CP efficiency. But for deeply inserted peptides, the efficiency should not be much lower. Therefore, as detailed in the point above, we have now clarified the states of peptides, with CP mainly detecting the well-inserted peptides, DP probing both the loosely bound or partially dissolved peptides, and INEPT selecting the dissolved phase.

We have also rephrased and expanded the original sentence: “At 298 K, the well-inserted peptides are relatively rare as evidenced by the 8 times stronger peptide signals in DP than in CP spectrum. The peptide peaks at 40-60 ppm region shows identical ^{13}C chemical shifts in INEPT and DP spectrum but change slightly from those in CP spectrum. Taken together, the hBD-3 analog undergoes minor structural changes upon binding to membrane, whereas there is little difference between the dissolved state and the state with loose attachment to membrane.”

4) At the line 105, at the temperature 269K, could the author comment if the sample is frozen? If it is, have the author considered that the peptide in the frozen solution could cause strong CP signal? I would recommend a contrast experiment in which only peptide solution is measured at this temperature to make sure that it doesn't contribute to the observed CP signal.

We explored the temperature profile near the freezing point before systematic measurements. It turned out that at 269 K the sample is not frozen since the ^1H peak of water is sharp and dominantly high (Fig. S1).

5) Could the authors provide an estimate of the error on the distance fitting in Figure.

The ^1H spin diffusion technique is not an accurate distance-measurement method like REDOR. We estimate the error margin of distance fitting to be in general 1-2 Å. Therefore, we emphasize that the method is semi-quantitative and is used to estimate the distance.

6) line 215-218, the authors claimed that a higher lateral diffusion indicate a higher coerture and smaller vesicles and hBD-3 analog fragmented the lipid bilayer into smaller, ellipsoidal vesicles. Without further evidence, like EM images, I think the claim is bold.

We thank the reviewer for the insightful comment. We have now added TEM figures of the peptide-containing and peptide-free samples as Supplementary Fig. 6 to support the NMR observation.

Reviewer #3 (Remarks to the Author):

The manuscript by Kang et al report on site-specific structure and dynamics of human β -defensins in lipid bilayers by solid-state NMR experiments and molecular dynamic simulation. This study provides atomic level structural information crucial for understanding structure-function relationship of β -defensin and its effect in membrane-disruption. The β -defensin is highly polymorphic and retained the beta sheet structure when bound to membranes based on the ^{13}C - ^{13}C NMR experiments. Furthermore, the residue-specific insertion depths of β -defensin were determined by fitting the build-up curves of ^1H - ^{13}C correlated signals, and conformational mobilities of each residue were identified by dipolar coupling measurements. The phosphorus chemical shift anisotropy measurements indicate that the lipid phosphate headgroups were rigidified and the reorientation rates of lipids were extracted from 2D ^{31}P - ^{31}P spectra with different mixing time. The results and conclusions of different NMR experiments are quite consistent, and in combination with MD simulation the binding mode of β -defensin peptide to lipid bilayers is revealed. The investigation strategy is impressive, with the NMR data well interpreted and clearly presented, and the conclusions well validated. More importantly, the investigators employed and present a systematic approach to investigate protein dynamics and protein-lipid/protein-protein interactions, and the integrated solid-state NMR method could be applied to a wide range of biological systems. There are multiple novelties in the experimental design, including the idea of applying specific isotope-labeling schemes and 2D ^{31}P - ^{31}P exchange solid-state NMR experiments. Overall the manuscript is excellent and well written, and will be important to both biology community and solid-state NMR community.

We thank the reviewer for the positive comments of the experimental design and potential impact of this study.

There are a few comments that may require minor revisions and more explanations from the authors.

^{13}C spectra of mobile and rigid phase

While I highly agree with the site-specific data analysis of the 2D C-C spectra at different temperatures, I wonder how one can demonstrate that the mobile phase and rigid phase corresponds to dissolved and membrane-bound states, respectively. An explanation for the relationship between these concepts will help readers to clearly understand the data interpretation as the mobile/rigid phases and dissolved/membrane-bound states are frequently mentioned. For this biological system, the changes in spectral resolution and sensitivity may arise from several aspects: i) overall temperature dependent dynamics; ii)

changes of insertion depth and membrane-bound state, which affects various level of dynamics changes in different segments. Consequently, is it possible that at 298K the molecules are mobile and homogeneous but are still bound to membranes instead of being in dissolved state?

We thank the reviewer for the helpful suggestions. We have substantially expanded the second paragraph in the Results section to clarify the potential states of peptides and better explain the spectroscopic results from different polarization techniques:

“With 65-70 wt% hydration, the hBD-3 analog exists in three major states with decreasing mobility: dissolved in solution, loosely associated with the membrane surface, and deeply inserted into the lipid bilayers. The highly mobile hBD-3 analogs dissolved in the aqueous phase is detected through the J-coupling-based refocused Insensitive Nuclei Enhanced by Polarization Transfer (INEPT) technique^{36, 37}. ¹³C direct polarization (DP) with a short recycle delay preferentially selects the relatively mobile components^{38, 39}, with contributions from dissolved or loosely bound peptides, while ¹H-¹³C cross-polarization (CP) detects the peptides rigidified by their insertion in lipid bilayers. At 298 K, the membrane inserted peptides are relatively rare as evidenced by the 8 times stronger peptide signals in DP than in CP spectrum. The peptide peaks at 40-60 ppm region shows identical ¹³C chemical shifts in INEPT and DP spectrum but change slightly from those in CP spectrum. Taken together, the hBD-3 analog undergoes minor structural changes upon binding to membrane, whereas there is little difference between the dissolved state and the state with loose attachment to membrane.”

We agreed with the reviewer that multiple factors would cause changes in spectral resolution and sensitivity. We also agreed that at 298K, hbd-3 analog may still be bound to membrane, as evidenced by peaks from cross polarization. However, the peaks were too weak for analysis. Therefore, we had to decrease the temperature to 269K to achieve better resolution and sensitivity. It should be noted that even at moderately low temperature, it still needs 1920 number of scans for measuring a 2D spectrum, thus high-temperature experiments become almost impractical. Considering that the membrane is still in liquid crystalline phase, the moderate temperature decrease may induce trivial changes in the insertion depth, which would not affect the main conclusion of the paper.

Methods: solid-state NMR experiments

The experiments were well described, but some important parameters were absent for multiple NMR experiments (including some listed below but not limited to). Please also indicate whether the 1H decoupling was applied during the 31P experiments. Although the paper will be presented to general audiences, these experimental parameters are important for researchers to reproduce the same type of NMR experiments.

We thank the reviewer for the valuable advice. We have included the parameters and details of the NMR experiments in the Methods section. TPPM decoupling sequence and 1H decoupling field strengths of 50 kHz were used in ³¹P experiments.

Conclusions

At the end of discussion, it will be helpful to remind readers the key findings by reorganizing and integrating the conclusions of each section.

We appreciate the insightful advice and we have now restructured the conclusion paragraph to summarize and restate the key findings with deeper understandings and outlook.

Minor:

Page 3, line 50: “The solution or crystallographic structure of five β -defensins...” should be “...solution or crystallographic structures...”

Thanks for the comment, and we have corrected the writing.

Page 4, line 67: “Here we integrate 2D ^{13}C - ^{13}C , ^{13}C - ^1H , and ^{31}P - ^{31}P correlation solid-state NMR...” should be “... ^{31}P - ^{31}P correlated...”

We apologize for the grammar mistakes. We have corrected them as the reviewer suggested. We also have proofread the manuscript and corrected the spotted typos, spacing issues, and grammar mistakes.

Page 6, line 101: the values of NMR experimental parameters should be described in detail for both 1D and 2D ^{13}C experiments either in line or in experimental section. In particular, the duration of recycle delay and whether it is 90 degree or small angle excitation for ^{13}C DP experiment.

We have included more experimental parameter details both in the Solid-state NMR experiments part of the Methods section: “The typical recycle delays were 1.5-2.0 s..... DP using a 90 degree ^{13}C pulse with a short recycle delay of 2 s was used.....” We have also included more technical details for the ^{31}P experiments.

Page 7, line 133: “...suggesting this region has high conformational flexibility...” the observations more likely suggest the existence of conformers or conformational heterogeneity. It is not very convincing that these regions are highly flexible based on the changes in secondary structure.

We fully agree that it is a weak and distractive statement and we have deleted this statement as reviewer suggested.

Page 14, line 280: “...reveal the function-relevant structure of hBDs in lipid bilayers and its effects on membrane morphology.” The significance of this work also includes the function-relevant dynamics. The statement can be revised as “the function-relevant structure and dynamics” throughout the manuscript.

We have changed the statement to “the function-relevant structure and dynamics”.

Page 20, line 400: It is recommended to indicate the parameters such as recycle delay and RF field strength of ^{31}P pulses regarding to ^{31}P NMR experiments.

We have added more experimental parameter details used in the ^{31}P NMR experiments in the Methods section: “In all ^{31}P experiments, the radio frequency field strength of 50 kHz for ^{31}P was used with recycle delays of 1.5 s. Two-pulse phase-modulated (TPPM)

decoupling sequence with ^1H decoupling field strength of 50 kHz was used in both 1D static and MAS ^{31}P spectra.”

Reviewer #4

Kang et al present a nice study of the structure, dynamics, and interaction of defensin with membranes by ssNMR and MD, which improves our understanding of the functionality of this antimicrobial peptide. The authors discussed the morphology, polymorphism, and dynamics of the peptide in both “dissolved” and rigidified forms. More importantly, they determined the topology and binding interfaces of the peptide with the membrane combining ssNMR and MD observations. They also shed light on how the peptide modulates the membrane structure. Their results provide great insights into the interaction of defensin and membrane with high-resolution.

We thank the reviewer for the positive comments of the significance of this work.

Major questions:

1. Have you done J coupling-based experiments to distinguish the mobile phase? In DP spectra both mobile and rigid components show up. DP being more sensitive than CP suggests the presence of mobile components whereas DARR is not the experiment for their detection. Signals shown in DARR suggest somewhat rigidity for dipolar coupling signal transfer, therefore residues shown in Fig. 1c (298 K) can't be dissolved to retain sufficient dipolar coupling transfer. J coupling-based experiments need to be done for the detection of mobile components.

We measured the J coupling-based INEPT experiment for membrane-containing hbd3 analog samples to detect the dissolved phase. In J coupling-based spectra, we observed several resolved peptide peaks, whose chemical shifts matched those of DP spectra. We have now added the spectra in Fig. 1b.

We fully agree with the reviewer's critical comment. We have now rewritten the second paragraph of the Results section. It is not rigorous enough to directly label the signals in DARR spectra as dissolved peptides. DP DARR has been used to probe loosely bound or partially dissolved protein domains with low transferring efficiency, and we have added two references, 38 and 39. Accordingly, we have corrected the statement of the peptide states accordingly, with INEPT showing the dissolved peptides, DP showing the combined signal from partially dissolved or loosely bound peptides, and CP reporting the rigid, membrane-inserted phase.

2. Can the authors perform further experiments to determine the oligomeric state of the membrane-bound peptide?

We thank the reviewer for providing very helpful insight suggestion. We indeed have a plan to systematically study the oligomeric state of this membrane-bound peptide with multiple strategies, including 1) mixing peptides with different isotope-labeling schemes to form hybrid oligomer for observing inter-molecular interaction and 2) introducing ^{19}F -labeled residues by mutagenesis at critical position and using ^{19}F -CODEX experiment to determine the oligomeric state and inter-peptide distances. Such

investigations require the synthesis of many peptides with different labeling schemes and extensive spectroscopic efforts and will be the followup study.

3. Does the polymorphism of the peptide have anything to do with its oligomeric state?

We couldn't correlate the polymorphism of the peptide with its oligomeric state based on our current data.

4. Can the authors provide other measurements to show the perturbation of membrane morphology, such as DLS? It is necessary to show hBD is functional active under experimental conditions.

We have measured TEM to show the peptide-perturbed morphology of the membrane and have added the results as Supplementary Fig. 6. Also, the 1D and 2D ³¹P experiments provide strong evidence for membrane morphology perturbation. Furthermore, the direct observation of the membrane-bound state of hBD-3 already suggested hBD-3 was active under the experimental conditions.

Minor questions:

1. To quantitatively evaluate the dissolved and bound peptides, do you compare peak height or volume in DP and CP spectra? I wonder why there are dissolved peptide left in the pellet after centrifugation.

We have indeed systematically analyzed the ratio of well-resolved peak intensity in DP and CP spectra at 298 K. Typically, the peak height of DP spectrum are 8 times higher than those in CP spectrum.

Although hbd-3 analog can insert into membranes, the analog peptide itself is highly soluble. In our experiment, ~10 mg of hbd-3 peptide can be fully dissolved with only 400 μ L HEPES buffer, which equals to the concentration of ~5 mM. Centrifugation could not completely remove the water from the sample and the hydration level of the sample is high, 65-70% by estimation. Therefore, it is not surprising to have hbd-3 analog remaining in dissolved state in the well hydrated pellet. We have added the hydration level and clarified the peptide states in the second paragraph of the Results section.

2. If the peptide is dissolved in solution, will you expect to see peaks in a DARR experiment? Could the dissolved form loosely tethered?

We thank the reviewer for the insightful comment, and we have now rephrased the description to "partially dissolved or loosely bound peptides" for the signals observed in DARR. As discussed above in Major Question 1, DP DARR has been used for investigating partially dissolved peptides and proteins but the polarization transfer efficiency will be substantially lower.

3. With longer spin diffusion times, do you see the decrease of diagonal peak intensity as with increases of off-diagonal peak intensities? As both diagonal and off-diagonal peaks change their intensity in the spin diffusion process, it makes sense to normalize according to sample amount rather than diagonal peak height. Also is there a particular reason for showing cross sections at 6 and -12 ppm?

We have now added a new figure, Supplementary Fig. 5 to analyze the cross sections of ^{31}P 2D spectra in different ways. In Fig. S5a, we normalize the cross sections by number of scans and we did observe the decrease of the diagonal peak intensity with the increases of off-diagonal peak intensities. In Fig. S5b, We normalize the spectra by sample amount instead of the diagonal peak intensity and the off-diagonal peaks are consistently higher in peptide-containing samples.

There is no particular reason for showing cross sections at 6 and -12 ppm. They are used to show the condition across the span of the ^{31}P spectra. The effect can be directly visualized from the 2D spectra. The 1D cross sections are used to provide a closer look.

Supplementary Figure 5. Cross sections of 2D static ^{31}P - ^{31}P exchange spectra. The POPC/G membranes without (black) and with (magenta) hBD-3 analog are compared. The cross sections are extracted from 6 ppm and -12 ppm to cover the range of the 2D spectra with mixing times of 1, 5, 100, and 400 ms. The intensity was scaled by **a**, number of scans and **b**, the samples amount ratio factor (0.75) obtained from the integral of 1D static ^{31}P spectrum.

4. What is the recoupling pulse power used in DIPSHIFT experiment? What is the number of t2 points and its increment?

We have added those experimental details in the “Solid-state NMR experiments” section: “...the transverse ^1H field strength is 83.3 kHz, converting to effective field strengths of 102 kHz. The ^{13}C π pulse is using a field strength of 62.5 kHz. The number of t2 points is 9 with an increment of 16.03 μs .”

5. Please correct typos such as I30->G31 and 518 Da (?) for IVLG on Page 16.

We apologized for the typos. We have corrected them.

REVIEWERS' COMMENTS:

Reviewer #1 (Remarks to the Author):

In the revised manuscript, the authors have carefully addressed my comments, thus I recommend to accept this manuscript.

Reviewer #2 (Remarks to the Author):

The authors have addressed my major concerns in this paper. I am pleased with the modifications.

Reviewer #3 (Remarks to the Author):

The rebuttal letter from the authors have clearly addressed my questions and comments. In the revised manuscript, the results and conclusions are consistent and validated, with additional results and figures implemented in the supplementary materials. The manuscript is well suited for publication after revisions.

Reviewer #4 (Remarks to the Author):

The authors have addressed all my concerns and the current version is in a good shape. No further questions need to be addressed. I recommend for acceptance for publication.